# 'You Owe It to Yourself, Everyone You Love and to Our Beleaguered NHS to Get Yourself Fit and Well': Weight Stigma in the British Media during the COVID-19 Pandemic—A Thematic Analysis

Camila Carbone-Moane [1],* and Andrew Guise [2]

1   Independent Researcher, London W2 6DR, UK
2   Department of Population Health Sciences, King's College London, London SE1 9RT, UK; andrew.guise@kcl.ac.uk
*   Correspondence: camilacarbonem@gmail.com

**Abstract:** The portrayal of obesity in the media can impact public health by guiding peoples' behaviours and furthering stigma. Individual responsibility for body weight along with negative portrayals of obesity have frequently dominated UK media discourses on obesity. This study aims to explore how the media has represented obesity during the COVID-19 pandemic through a thematic analysis of 95 UK online newspaper articles published in *The Sun*, *The Mail Online*, and *The Guardian*. The first theme, lifestyle recommendations, accounts for media coverage providing 'expert' advice on losing weight. The second theme, individual responsibility, emphasises media appeals to self-governance to tackle obesity and protect the NHS during the pandemic. The third theme, actors of change, explores how celebrities and politicians are presented as examples of weight management. These results suggest that individuals are held responsible for their weight and accountable for protecting the NHS during the COVID-19 pandemic. Stigma can be furthered by the decontextualisation of lifestyle recommendations and exacerbated by the actors of change presented: Celebrity profiles reveal gendered goals for weight management, and politicians exemplify self-governance, which consolidates their power. In conclusion, individualising and stigmatising discourses around obesity have taken new forms during the pandemic that link health responsibility to protecting the NHS and invokes celebrities and politicians to foster action.

**Keywords:** obesity; stigma; COVID-19; online news; media analysis

## 1. Introduction

By defining an issue and shaping perceptions, the media can play a powerful role in creating and perpetuating particular understandings of obesity (Buse et al. 2012). The portrayal of obesity in the media has the potential to affect public health positively or negatively, as described for a plethora of health issues. For example, when the human papillomavirus immunisation was introduced in the UK, positive coverage in newspapers likely increased acceptability for some, but a focus on the vaccine as a 'promotor' of promiscuity was detrimental to public health strategies promoting its uptake (Hilton et al. 2010). Similarly, the depiction of dieting and cooking as "unmasculine" in UK newspapers has been understood to undermine healthy food practices (Gough 2007).

The medicalisation of excess weight has prompted it to be regarded as a disease and located the problem within unhealthy or 'not normal' individuals (Stefánsdóttir 2020). UK Government strategies and policies—as well as governments globally—have followed this individual framing and demanded high individual agency to reduce weight (Ulijaszek and McLennan 2016; Theis and White 2021). The UK National Health Service's (NHS) long-term plan focuses on the individual and frames obesity as a burden for the health system (Flint 2020b). Such an individualistic framing leads to moral judgment and perpetuates

stigma (Cotter et al. 2021), including through shaping patient–provider interactions and jeopardising the mental and metabolic health of individuals (Tomiyama et al. 2018). The stigma that results from health-promoting discourse is then a concern for public health responses and the need to prevent or mitigate this stigma (Bayer 2008).

Public health has increasingly focused on risk factors and encouraged modes of health surveillance that place responsibility on the individual for their health through emphasis on diet and exercise (Armstrong 1995). When weight is presented as a result of self-care in such discourses, then regulated lifestyle choices lead to self-fulfilment (Veinot 2010), and failure to self-manage weight is regarded as deviance (Shilling 2002). Following Goffman (1963), body weight and its management becomes both the physical and social attribute that can devalue an individual's identity and hinders full social acceptance. Tomiyama et al. (2018, p. 1) define weight stigma 'as the social rejection and devaluation that accrues to those who do not comply with prevailing social norms of adequate body weight and shape'. A failure to fulfil society's expectations of weight and self-regulation can lead people to internalise negative understandings of self (Williams and Annandale 2019), generating psychological distress, social anxiety (Ratcliffe and Ellison 2013), and unhealthy eating and exercise avoidance (Täuber et al. 2018).

Terminology relating to weight is key to preventing furthering stigma and its detrimental consequences. The term 'fatness' has been reclaimed by the National Association to Advance Fat Acceptance (NAAFA 2015), a fat acceptance organisation, because terminology such as 'person with obesity' can be seen to medicalise weight and declare its excess as a disease. Meadows and Daníelsdóttir (2016) warn that use of the term 'obesity' may attribute a characteristic that could produce a judgment. However, 'fat' has been regarded as offensive in America by parents (Knierim et al. 2015) and university students, who preferred to describe their own and someone else's weight as 'overweight' and 'unhealthy' (Trainer et al. 2015). Consensus on non-stigmatising terminology is then challenging. The lack of consensus on weight terminology was addressed by a collaboration between stakeholders from Obesity UK, health professionals, researchers and representatives from National Health Service England Diabetes and Obesity in the UK (Albury et al. 2020). The negative connotations of the term 'obese' was agreed on, while the use of person-centred language ('people with obesity') and of language free from judgement or negative connotations was recommended. Whilst acknowledging the tensions around terminology, and especially the risk of medicalising some experiences, the present article will follow this guidance. The present article will only use the term 'obese' when quoting from newspaper articles being analysed. The use of person-centred language—'people with obesity—will be maintained throughout the manuscript, and an evidence-based approach when discussing experiences of obesity will aim to avoid assumptions or judgement arising from the text.

The media serves as a platform that can reinforce stigma (Maiorano et al. 2017). Anti-fat attitudes have been perpetuated through stigmatising portrayals of body weight and its management. Such media practices include use of moralising addiction metaphors to describe excess body weight (Cotter et al. 2021), and charicatured portrayals that reinforce stereotypes involving laziness (Flint et al. 2016) across UK newspapers. Similarly, North American media practices have consisted of blaming individuals for a lack of will power or moral fortitude for their excess weight (Cain et al. 2017), presenting images that dehumanise by focusing on lower bodies and excluding the faces of people with obesity (Heuer et al. 2011) and reinforcing stereotypes of laziness and irresponsibility (Glenn et al. 2012). Previous findings include the framing of obesity as an individual responsibility, recently demonstrated in British and German online newspapers (Atanasova and Koteyko 2017). Weight stigma can also be gendered. American news media representations of women's exercise have emphasised beauty as the aim of weight management (McGannon and Spence 2012), while UK newspapers reinforced dieting as a "feminine" practice (Gough 2007), with both creating particular norms that in turn generate stigma and the rejection of other experiences. However, coverage on the societal causes and

solutions of obesity has increased in American network news (Gearhart et al. 2012) and UK newspapers (Hilton et al. 2012).

The production of weight stigma in the media can be understood as a 'constantly changing social process' in relation to power and social inequality (Parker and Aggleton 2003, p. 14). Following this, COVID-19 has shifted many aspects of daily life, especially in relation to health, as well as exacerbating or drawing attention to social inequalities. The COVID-19 pandemic has been the focus of extensive media reporting, especially on obesity, which was rapidly confirmed as a primary 'risk factor' for middle-aged adults in the UK. Epidemiological analysis—widely reported in the media—has demonstrated that the odds of having COVID-19 were 55% greater for patients with obesity (Yates et al. 2020). Excess weight was also associated with COVID-19 hospitalisation and death (Public Health England 2020). The potential mechanisms for any link are complex. Increased adipose tissue may impair the response to a viral disease (Singla et al. 2010). Obesity may impact intensive therapy care and disease progression by complicating ventilation and rehabilitation (Sattar et al. 2020). However, there is inconsistency about COVID-19 outcomes and obesity, and given the methodological inconsistencies, more prospective studies are needed to reduce bias and fully understand such links (Colombera Peres et al. 2020). Furthermore, whilst obesity may be understood as a 'risk factor', broader analyses point to the role of social inequalities generally (Public Health England 2020) in COVID-19 risk disparities in the UK: Black ethnic groups, those living in deprived areas, and those with high-risk jobs had higher COVID-19 diagnosis rates. Most deprived areas (UK Parliament 2021) and Black adults (UK Government 2021) have a higher prevalence of excess weight. Whilst obesity and COVID-19 can then be linked, with obesity presented as a 'risk factor', this needs to be understood within broader processes that generate social inequalities, and through which stigma might be produced.

Media reporting on the COVID-19 epidemiology of obesity could then generate stigma and impact on public health. Sociological observations of everyday life and social media have revealed that the COVID-19 pandemic led to a moral imperative of health, as weight and lifestyle anxieties increased (Monaghan 2020). Pearl (2020) commented on the implications of weight stigma on social media during lockdown and provided examples such as the 'before and after quarantine' memes depicting how people gained weight, a theme also recognised by Aslan's (2021) discourse analysis of COVID-19 internet memes that revealed physical appearances as a focus for quarantine humour. Additionally, Branley-Bell and Talbot (2020) identified social media posts as a source of anxiety for individuals with experience of eating disorders in the UK by developing and implementing a mixed methods online survey. Flint (2020a) analysed three media sources that presented obesity as a societal burden and told individuals to self-manage their own weight. Brookes (2021), in a similar vein, examined the British press during lockdown using a novel software-driven approach to recognise linguistics patterns based on keywords that revealed obesity-stigmatising discourses that constructed people with obesity in fatalistic terms, alongside an increase in race-related health disparities relating to obesity. These initial analyses of media coverage during COVID-19 suggest a potential for new meanings and media discourses with impacts on obesity and public health responses. This study aimed to develop closer examination of what these discourses are, as well as identify others.

## 2. Materials and Methods

The research design was a thematic analysis building on media discourse analysis (Braun and Clarke 2006) of online newspaper articles published during the COVID-19 pandemic. The design builds on discourse analysis used in the analysis of health topics in newspapers and magazines (Gough 2007; Glenn et al. 2012; McGannon and Spence 2012; McGannon et al. 2016) to understand obesity representations during the COVID-19 pandemic (Cheek 2004; Willig 2000). Thematic analysis techniques are suitable to identify patterns and nuances within the media discourse and to incorporate techniques from discourse analysis and grounded theory such as comparative strategies (Braun and Clarke 2006).

UK newspapers were sampled to provide diversity in political alignment and readership profile (MacArthur and Reeves 2019; Seale et al. 2007; Wells and Caraher 2014). Online news media were sampled from *The Mail Online*, *The Sun*, and *The Guardian* to achieve a range in politics and readership demographics. *The Mail Online* usually supports the right-wing Conservative party, while *The Guardian* is more aligned with the Labour party (Wells and Caraher 2014). According to the National Readership Survey (2017), *The Guardian* is the most read daily online from the 'quality' category, *The Mail Online* from the 'mid-market', and *The Sun* from 'popular' newspapers. In addition, *The Mail Online* has the highest reach in adults over 35 years and ABC1, while *The Sun* has it in adults 15–34 years old per month and C2DE (National Readership Survey 2017).

Newspaper articles were obtained using the Nexis electronic database (Strong and Wells 2020; Wells and Caraher 2014; Seale et al. 2007). The time frame considered started in January 2020 and ended in May 2020. This timeline was in response to easement of social restrictions in June in the UK and based on judgement of what volume of data could be analysed in the time available to the research team. The consequent timeline of lockdowns and restrictions means that the present article provides a snapshot of the initial weight-based discourse that occurred during the early stages of the ongoing pandemic. Articles were identified using the following search terms: 'coronavirus', 'covid', 'overweight', 'fat' and 'obes!'. The search yielded 578 articles: 215 from *The Mail Online*, 214 from *The Guardian*, and 149 from *The Sun*. They were then sifted for relevance as some articles contained the search terms 'fat' and 'covid' but were unrelated to obesity. After sifting, 186 relevant articles were obtained: 62 from *The Mail Online*, 64 from *The Guardian*, and 60 from *The Sun*. They were transferred to NVivo, where 95 were analysed until no new themes arose and the sample equally represented each newspaper, resulting in 31 from *The Mail Online*, 32 from *The Guardian*, and 32 from *The Sun*.

Braun and Clarke's (2006) approach to thematic analysis informed data analysis. First, articles were read to gain familiarization with their content. Second, 10 articles were coded line by line using NVivo. Initial themes were identified. Third, codes were explored and compared for overarching themes. Grounded theory's concept of constant comparison was used to merge and divide codes into themes. It aided in categorising data and producing codes categories (Glaser and Strauss 2006) by diagramming relationships between initial themes. Thus, initial codes began to be grouped into overarching themes. Memos were hand-written throughout coding to guide analysis by connecting themes (Glaser and Strauss 2006). Furthermore, colour-coded mind maps were created to group subthemes.

## 3. Results

The results focus on three themes that in combination describe an overarching theme of how obesity was principally represented as a rationale for modifying lifestyles during the pandemic to protect against COVID-19. The first theme, lifestyle recommendations, accounts for media coverage providing 'expert' advice on how to lose weight. The second theme, individual responsibility, appeals to self-governance in order to tackle obesity and protect the NHS during the pandemic. The third theme, actors of change, explores how celebrities and politicians are presented as examples of weight management.

### 3.1. Lifestyle Recommendations

Lifestyle recommendations were a key focus for media coverage of obesity and COVID-19, often highlighting the need to improve diet and increase exercise to maintain a healthy weight; this was featured in 19 articles (12 from *The Sun*, 6 from *The Mail Online*, and 1 from *The Guardian*). Exercise was mainly promoted in *The Sun*, while diet advice was equally present in *The Mail Online* and *The Sun*. The narrative structure employed by health, food, and sports reporters was dominated by quoting expert advice in 9 articles (7 from *The Sun*) with labels and titles given to frame the credibility of the advice. Thus, the medicalisation of the subject is reinforced with a focus on the weight-loss industry. In addition, a sense of urgency is created to adopt lifestyle recommendations. Advice given included avoiding

processed foods, going for a walk or run every day, or joining online fitness classes: ' . . . dietitian Helen Bond says the snacks you munch on during lockdown can make a huge difference to your waistline. She tells us: "Snacking can be part of a balanced diet . . . it's important to choose our snack food wisely" . . . ' (The Sun 2020f).

' . . . Tamara Willner, a nutritionist at British weight loss company Second Nature said . . . Opt for fresh vegetables, (e.g., spinach and peppers), minimally processed meat, fish, or vegetarian alternatives (e.g., chicken, salmon, tofu), and wholegrain carb options (e.g., brown rice or rye bread).' (Mail Online 2020e).

'Perhaps a new academic paper, COVID-19 . . . . by the scientist David C Nieman, might focus minds . . . Regular exercise also guards against obesity, which as Nieman makes clear 'markedly increases the risk for hypertension, type 2 diabetes, and cardiovascular disease, three of the most important underlying conditions for COVID-19'. (The Guardian 2020a).

Expert advice is specific, although feasibility can be an obstacle to following them because of food availability or social resources. For example, the consumption and preparation of specific foods are suggested in *The Mail Online* even though they may not be widely available or may be complicated to make without the prior knowledge, time, or skills available.

'Many Brits are low on selenium, which can put you at risk of infection . . . If you do not like fish, then nibble on some seaweed snacks instead (fish get their omega-3 from seaweed).' (Mail Online 2020d).

"We take turns to cook healthy Mediterranean-style meals, with lots of oily fish, nuts, vegetables and legumes. Clare has been making lots of fermented food, including sauerkraut and kimchi from ingredients such as cabbage, ginger and salt, while I have been producing home-made yoghurt which is rich in bacteria." (Mail Online 2020b).

### 3.2. Individual Responsibility

Individual responsibility to tackle obesity and protect the NHS during the COVID-19 pandemic was present in 21 articles: 11 from *The Sun*, 8 from *The Mail Online*, and 1 from *The Guardian*. The second person singular ('you') was employed by media discourses to address readers directly in 15 articles—mainly in *The Sun* (*n* = 7) and *The Mail Online* (*n* = 7). Only two articles in *The Sun* proposed measures to tackle obesity at a societal level: bariatric surgery and vehicle emission taxes. The following quotes repeatedly use 'you' to emphasise individual responsibility and also reinforce the personification of obesity by using 'being obese' instead of 'having' obesity (Katz 2014). The first quote from *The Sun* fuels obesity fear by referring to a slow death, emphasises social responsibility by mentioning loved ones, and refers to weakened health services. The second quote from *The Sun* describes how someone with excess weight burdens doctors as they are considered a 'practical nightmare' to manage. The third from *The Mail Online* highlights the effect of excess weight on lungs but does not reference death directly.

'If you are seriously overweight, you have to change your mindset, your attitude and your lifestyle . . . Overeating and not exercising is serious and will be slowly killing you . . . You owe it to yourself, everyone you love and to our beleaguered NHS to get yourself fit and well.' (The Sun 2020e). "The extra weight creates problems with breathing, clogs up your arteries and makes you a practical nightmare for doctors to shift, scan and operate on, according to the World Obesity Federation." (The Sun 2020a).

'One reason is the more overweight you are, the lower your lung capacity. So if COVID-19 attacks your lungs, then you are more likely to end up in intensive care.' (Mail Online 2020d). Contrarily, *The Guardian*'s discourse did not use the word 'you' or the personification of obesity, but it did underline in the one article that emphasised the individual responsibility that health services should be protected by an active lifestyle.

"Yet the science is crystal clear: exercising for 30 min every day is a far better way to help the NHS than clapping for two minutes every Thursday." (The Guardian 2020a).

### 3.3. Actors of Change

During the pandemic, celebrities and politicians were presented as particular 'actors of change' that exemplified the enactment of individual-focused lifestyle recommendations to control weight. This was evident in 31 articles: 15 from *The Sun*, 10 from *The Mail Online*, and 6 from *The Guardian*. Celebrities were role models during the COVID-19 pandemic, and 10 articles provided further recommendations based on their lockdown weight loss experiences.

Seven articles presented male celebrities taking responsibility for their own health in *The Sun* and *The Mail Online* but not in *The Guardian*. For example, *The Mail Online* celebrated the 'Hairy Bikers' (Si King and Dave Myres), British television chefs, as they transformed their eating habits after a health scare. Even though the article recognizes them as pioneers in tackling men's complicated relationship with food—which is then explained to be characteristic of women—a particular understanding of masculinity is protected by adding to the reader 'don't admit it' and highlighting their careless attitude:

' . . . and how we loved their laissez-faire attitude to everything—waistlines included . . . they took action they knew would lessen the chances of their lifestyle heading straight to diabetes, strokes, heart attacks and who knows what else . . . The Hairy Biker diet approach was the first to tackle the fact that men can have just as complex relationships with food as women do, but don't admit it.' (Mail Online 2020c).

Similarly, *The Sun* described Ricky Hatton, British professional boxer, as seizing the lockdown opportunity to lose weight by avoiding alcohol and fast food. As a result, he regained his strength and willingness to fight again and so restoring a particular identity and reinforcing particular understandings of masculinity:

"Ricky Hatton is fighting fit after boredom from the coronavirus lockdown pushed him to get back into top shape—now he wants to knock someone out . . . And now the 41-year-old Mancunian has admitted he would relish a ring return after giving up the beer and takeways to lose weight and make sure he is in his best condition for ten years." (The Sun 2020c).

In contrast, female celebrities were presented only in three articles—two from *The Sun* and one from *The Guardian*. One article in *The Sun* focused on how these women maintained their fitness during lockdown by eating home-cooked meals and working out with family. Compared to men, no emphasis was given on mitigating COVID-19 or comorbidities. Only once is a motive is given: low self-esteem. The pun used for the title ('Locking good') and the descriptions of posting their toned figures on social media prioritises the aesthetic role of their body over their health:

'Chloe Ferry—In a post on Instagram showing her body transformation, the reality TV favourite told her 3.4 m followers that a bout of low self-esteem prompted her to start going to the gym, something which was also 'good' for her mind. Since the pandemic began, she has been eating a high-protein breakfast and doing home workouts . . . '. (The Sun 2020d).

Another article from *The Sun* praised the weight loss of Adele, a British singer. In contrast to the previous example, the weight loss is presented as the reason for no longer being in the COVID-19 risk group. However, unlike male celebrities, the aesthetic role of her body continues to be a priority as a female celebrity:

"Look at superstar Adele. She has gradually lost around seven stone and is now far healthier, fitter and thank-fully no longer in the at-risk group when it comes to COVID-19. Adele was a beautiful-looking woman before her dramatic weight loss and she's still utterly gorgeous." (The Sun 2020e). Male politicians are also examples of lifestyle change during the COVID-19 pandemic in 21 articles—mainly from *The Sun* ($n = 10$) and less so in *The Mail Online* ($n = 6$) and *The Guardian* ($n = 5$). British Prime Minister Boris Johnson, who was hospitalised with COVID-19, is presented as acknowledging his excess weight and blaming it for his hospitalisation in 18 articles. He is continuously depicted as going for runs and doing exercise during his recovery process in *The Sun* and *The Mail Online*. Thus, his authority is protected through representations allowing for individual responsibility and agency in trying to self-mitigate his weight through a healthy lifestyle. The following

quotes highlight how he sets an example for the nation and wishes to tackle COVID-19 and obesity in tandem:

' . . . .Boris last month spent three days in intensive care fighting off the virus after his symptoms failed to clear following a ten-day spell in self-isolation . . . .He is now convinced that his weight is the reason the virus . . . .The PM has often been photographed jogging or riding a bike' (The Sun 2020b).

'Today was the second in a row Mr Johnson has taken a morning run after he reportedly decided his touch-and-go battle with COVID-19 . . . ' (Mail Online 2020a).

'Labour has hailed a "welcome conversion" by Boris Johnson as Downing Street confirmed that the prime minister hopes to lead a public health drive, having blaming his stint in intensive care on obesity . . . ' (The Guardian 2020b).

## 4. Discussion

This analysis has described in-depth UK media discourses of obesity deployed during the COVID-19 pandemic. The themes reported focus on how lifestyle recommendations were prevalent and relied on expert advice. Individual responsibility was also central to messaging, particularly with individuals called to protect the NHS. Finally, lifestyle recommendations and individual responsibility were exemplified involving particular gender norms, with celebrities and politicians as particular actors of change. These results suggest media discourses on obesity continue past processes and representations of stigma but have also taken on new forms during the pandemic and may have particular implications for public health.

Differences were identified between the newspaper outlets analysed and patterns can suggest a link to the newspaper characteristics identified in the National Readership Survey (2017), *The Sun* covered all themes more frequently, with a slight emphasis in exercise over diet lifestyle recommendations and male actors of change. This might reflect it having the highest reach in younger adults and being described as a 'popular' outlet. *The Mail Online* was categorised as a 'mid-market' newspaper, so its content followed similar patterns as *The Sun*, but they were present in about a half of analysed articles from *The Mail Online* compared to *The Sun*. It did not mention the need to protect health services in the articles analysed and did not present female examples of actors of change. On the other hand, *The Guardian* had the least number of articles analysed that presented these themes. As the most read in the 'quality' category, it presented factual information without providing diet advice or examples of celebrities as actors of change. Individual responsibility was only once underlined when giving exercise advice to help health services. It did not engage in employing the use of 'you' or the personification of obesity as *The Sun* and *The Mail Online* did, suggesting a more neutral language. In fact, sensational language has the potential to shape health behaviours as observed in the use of misleading superlatives in stem cell therapy coverage (Pham et al. 2021).

Our findings build on past studies which show how obesity, and health generally, are commonly presented in media discourses as an individual obligation. Such a framing has been evidenced before in UK government strategies and policies (Ulijaszek and McLennan 2016; Theis and White 2021). Obesity has also been attributed to individual governance before in UK media (Flint et al. 2016; Atanasova and Koteyko 2017; Cain et al. 2017), and health has been defined in British media as the result of self-surveillance (Neresini et al. 2019). The analysis, though, suggests that media discourses during COVID-19 have taken on particular novel forms that accentuate this idea of individual responsibility: Individuals are held accountable for protecting the NHS during the COVID-19 pandemic, and in so doing, demarcating new ways in which individuals bear responsibility for societal challenges. This novel form has also been reported in the British press coverage of COVID-19 by Brookes (2021) by identifying themes through keywords frequencies and providing extracts from various UK newspapers. The present article adds a more in-depth analysis of those themes and how they manifest in different media sources. Similarly, individuals are called to protect the NHS from the obesity burden on health professionals and costs

in the 'Tackling obesity' government policy paper: 'And we owe it to the NHS to move towards a healthier weight . . . . you can play your part to protect the NHS and save lives' (Government UK 2020). Individual responsibility was also drawn in official UK press releases during the pandemic urging the population to 'lose weight to beat COVID-19 and protect the NHS' (The Lancet Diabetes and Endocrinology 2020). The moral imperative to 'protect the NHS' during the COVID-19 pandemic becomes another potential means by which stigma is enforced and people internalise negative self-image, whilst citizens with a healthy body are legitimised and considered morally superior for how they are protecting society (LeBesco 2011; Link and Phelan 2014). These findings then add to the body of literature on how the NHS is historically framed as being burdened by obesity (Bivins 2020). However, during a crisis or periods of change, protecting the NHS can take on additional meanings, such as the revival of imperial national identities based on race and heritage in the context of Brexit (Fitzgerald et al. 2020). Protecting the NHS also implies an alignment to individualistic values which minimise the political reasons why the NHS was overburdened during the pandemic (Brookes 2021). As a result, we argue that weight self-management in the era of COVID-19 becomes a patriotic obligation to the NHS that creates particularly acute forms of weight stigma.

The lack of discussion of societal causes of obesity in the articles analysed further reinforces the importance of self-governance and negates the potential for broader structural interventions and policies to address weight. It echoes the stance taken in British and German newspapers (Atanasova and Koteyko 2017) and UK government policies and strategies (Ulijaszek and McLennan 2016; Theis and White 2021). However, it contradicts the recent increase in societal impact and solutions of obesity observed in American network news (Gearhart et al. 2012) and UK newspapers (Hilton et al. 2012). Evidence is increasingly accumulating that effective responses to obesity need to operate at multiple levels, including those that address social and structural barriers (Beauchamp et al. 2014; Backholer et al. 2014), for example, sugar taxes in the UK reduced purchases of high sugar drinks, but the purchase of sugar increased tremendously. Thus, the neglect of societal causes casts a shadow over the overall effect on lifestyle, health outcomes, and inequalities (Pell et al. 2021). Structural factors were neglected overall in the response to the COVID-19 crisis: Poorer people could not adhere to lifestyle guidance because of limited public space to exercise, and social distancing was not feasible in crowded homes (Drury et al. 2020). A disregard of social factors in media discourse then potentially exacerbates the social gradient of obesity and health generally. Caduff (2020) describes that extremes have been normalised during the COVID-19 pandemic, and pandemic measures exacerbate already existing inequalities for those marginalised. The analysis here suggests that this also applies to weight stigma and how it falls on particular groups within society.

Actors of change in the media were instrumental in exemplifying lifestyle recommendations and emphasising individual responsibility. Other literature has quantified politician's media coverage during the COVID-19 pandemic. Johnson's general COVID-19 coverage in online news sources across 11 countries was predominantly positive (Krawczyk et al. 2021), which resonates with COVID-19 and obesity coverage in UK online news (Brookes 2021). The role of celebrities has evolved from endorsements of processed food that increase obesity risks (Zhou et al. 2020) to the exposure of less discussed stigmatising health topics, for example, an online survey about Carrie Fisher's, American actress, mental health advocacy in the media revealed the potential to reduce stigma (Hoffner 2020). Our findings instead suggest the potential for less productive reframings of issues fostered by celebrities. The underlying implication of covering celebrities' individual stories is that broader societal causes are not discussed (Tyler and Slater 2018). Instead, celebrities and politicians fostered, and so reinforced, individualist framings of obesity, and so potentially undermining public health goals.

Furthermore, celebrities' stories exposed gendered differences and reinforced particular modes of self-governance of health. Gendered differences between celebrities exemplifying lifestyle changes during the pandemic coincide with past studies that outlined how

masculine identities are preserved by a disinterest in health and healthy food in American media (Courtenay 2000) and UK newspapers (Gough 2007). Coverage of celebrity female weight-loss was not directly related to risk from COVID-19 but to self-esteem, which can be linked to the culturally understood aesthetic purpose of slenderness for women (Tischner and Malson 2008). On the other hand, the coverage of celebrity male stories doubled compared to women's, perhaps to overcome the fact that a particular understanding of masculinity has been identified as a dieting barrier before (Gough 2007) and that male gender was associated with more severe COVID-19 outcomes (Gebhard et al. 2020). To our knowledge, no gender differences have previously been identified in obesity coverage during the pandemic. Therefore, our findings suggest that self-governance depicted in UK media legitimises and reinforces specific gender stereotypes and consequently could potentially construct other identities as deviant or morally inferior.

The representation of politicians adds further to this role for specific 'actors of change' of individual-focused responsibility. Self-governance is exemplified by Boris Johnson to align newspapers to government recommendations. His position as a moral citizen is legitimised because of his physical activity during recovery from COVID-19 (Täuber et al. 2018). Link and Phelan (2001) explain how stigma depends on social and political power: When negative attributes are attributed to someone, there is a status loss that exacerbates inequalities. Johnson's identity belongs to the dominant white male body, which is more likely to be in a position of power and so he can counteract weight stigma (Link and Phelan 2001). His power to counteract obesity discourses is established when he is not blamed for contracting COVID-19. Instead, he is framed as proactively self-managing it through exercise. Thus, obesity discourses of politicians exemplify self-governance and consolidate their particularly male power. The present social science research was timely and fills a knowledge gap in the portrayals of obesity in the media during the COVID-19 pandemic. It therefore provides evidence for the need to implement strategies that reduce weight bias in the media. Pearl (2018) proposes a media pledge to stop the use of stigmatising discourses in news content and public health messages by using counter-stereotypical images and reinforcing positive health campaigns, which elicit an increased motivation for healthy behaviours. World Obesity (2021), a global advocacy organisation, advocates for a fair portrayal of people with obesity and provides a positive image bank to encourage positive coverage. Kim and Willis (2007) suggest that an increase in the coverage of societal causes of obesity can aid in counterbalancing the discourse of obesity as an individual responsibility. The strategy to increase the coverage of weight stigma stories that educate on the harmful implications of placing responsibility on individuals (Puhl et al. 2017) could reduce weight bias in the media caused by the appeal to protect the NHS during the pandemic. Overall, further research on how to increase public support in the media and health campaigns without placing guilt or moralising health behaviours is needed (Pearl 2018).

The present study has limitations regarding the scope of data. First, only online news was included. Second, as the pandemic is ongoing, a snapshot from the onset until the end of the first lockdown in the UK was provided. Future research could explore how obesity representations varied throughout the pandemic according to the subsequent implementation of social restrictions and their corresponding easements. Furthermore, obesity framings in different media outlets could be further compared to understand more nuanced weight discourses during the pandemic.

## 5. Conclusions

In conclusion, individualising and stigmatising discourses around obesity have taken new forms during the COVID-19 pandemic that link health responsibility to protecting the NHS and invokes celebrities to foster action. The power of the media to shape public perception of obesity and COVID-19 can impact adherence to public health strategies. By recognising the particular media discourses being used and deployed, there is potential

for addressing weight stigma, and so public health goals could be aided and the uptake of recommendations would improve during a health crisis, such as the COVID-19 pandemic.

**Author Contributions:** Conceptualization, C.C.-M. and A.G.; methodology, C.C.-M. and A.G.; software, C.C.-M.; validation, C.C.-M. and A.G.; formal analysis, C.C.-M.; investigation, C.C.-M. and A.G.; resources, C.C.-M. and A.G.; data curation, C.C.-M.; writing—original draft preparation, C.C.-M.; writing—review and editing, C.C.-M. and A.G.; visualization, C.C.-M. and A.G.; supervision, A.G.; project administration, C.C.-M. and A.G.; funding acquisition, C.C.-M. and A.G. All authors have read and agreed to the published version of the manuscript.

**Funding:** This research received no external funding.

**Institutional Review Board Statement:** Not applicable.

**Informed Consent Statement:** Not applicable.

**Data Availability Statement:** No new data were created or analyzed in this study. Data sharing is not applicable to this article.

**Conflicts of Interest:** The authors declare no conflict of interest.

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
