# Peer review of "‘You Owe It to Yourself, Everyone You Love and to Our Beleaguered NHS to Get Yourself Fit and Well’: Weight Stigma in the British Media during the COVID-19 Pandemic—A Thematic Analysis"

_socsci, doi:10.3390/socsci10120478_

Round 1

Reviewer 1 Report

This paper reports on a content analysis of 95 news articles on obesity and covid-19 in the early months of the pandemic. With major revisions, this paper has the potential to make a significant contribution to the literature. The authors provide a thorough critique of news media discourses in their discussion section. The introduction section is, at times, a bit convoluted and would benefit from editing for clarity and organization. Further commends and suggestions for each section of the paper are provided below.

Introduction:

  • The reader may benefit from added clarity in the first two sentences of the introduction. What do the authors mean by ‘obesity representations’ and are the public health responses they influence positive, negative, or both. Perhaps some specific examples would be helpful here as well. The point made in the second sentence may also be somewhat lost on the reader, as the sentence is worded in a slightly awkward way. Perhaps something like “the medicalization of excess weight has led to obesity being declared a chronic disease…”
  • Line 29: I think the authors mean to say “as well as governments globally”
  • Lines 49-51: A direct quote by Tomiyama et al is provided here, but a page number is not given.
  • Lines 53-54: Perhaps swapping the order of phrases in this sentence will enhance clarity for the reader: “By defining an issue and shaping perceptions, the media can play a powerful role…”
  • The authors suggest that weight is associated with likelihood of testing positive for covid-19, citing Yates et al. 2020. It is important to consider whether or not greater likelihood of testing positive is associated with increased adipose tissue, per se, or if a third factor provides a better explanation (such as individuals with lower SES being more likely to be working in high-risk jobs, and therefore at higher risk for covid-19. We also know that low SES individuals also tend to have higher body weights). Critiques may also be made for covid-19 outcome research based on body weight. The authors are encouraged to present this research more tentatively in their introduction.
  • Given that Flint and Brookes, and other researchers, have commented on media covered throughout the pandemic, it is important to provide the reader with a review of these findings and to position this research within the literature. Further, the authors do not describe the purpose of the study in context, nor do they describe how the current student adds to the literature on weight stigma in the covid-19 pandemic.
  • There has also been previous discussion of quarantine-15 memes in the literature, including a content analysis. This research may be important to review in the introduction section

Methods:

  • The authors provide a rationale for the data collection period of January-May 2020, as easement of restrictions in the UK occurred in May 2020. Given the ebb and flow of lockdowns, the timing of the first published reports of obesity as associated with covid-19 outcomes, and the continued reporting of such findings after May 2020, further clarification of this narrow window would be appreciated. It may be that nearly 100 articles was deemed sufficient for analysis– however it may be important to note that this represents only a small snapshot of the overall weight-based discourse that occurred throughout the pandemic
  • The authors provide a thorough explanation of their process of thematic analysis in the current study
  • The paragraph on language usage and preferences seems out of place. This may be better in the introduction, with added context and connection to the study

Results:

  • Within the ‘lifestyle recommendations’ theme, it is unclear if this theme only relates to lifestyle recommendations made by experts within articles, or if other forms of lifestyle recommendations are included in this theme as well. For example, it is plausible that a journalist with a history of writing about health, nutrition, and/or physical activity would feel comfortable writing their own advice into articles. It is unclear in the examples provided in the results or explanation of the theme whether these kinds of recommendations were included.
  • There are different numbers of examples across themes. A reader may appreciate a few more examples for themes where only two are provided

Discussion:

  • Lines 268-271: The sentence beginning with “Finally, particular actors of change…” is confusing and it is difficult to understand the intentions of the authors
  • The authors provide an in-depth and critical discussion of the obesity discourses in the context of larger social issues, such as the social determinants of health
  • The discussion would benefit from the identification of potential solutions or strategies for reducing weight bias in news media coverage.
  • Limitations of the study or suggestions for future research are not made

Reviewer 2 Report

This study investigated themes related to obesity and weight management in the British media during the initial months of the COVID-19 pandemic. This study draws necessary attention to the ways in which media promotes weight stigma, particularly in light of COVID-19. The manuscript could benefit from additional development of the introduction and discussion, as detailed below.

Primary Comments:

The comment on page 2, lines 60-64 (beginning with “More recently”) could be better integrated with the rest of the introduction. I see that you’re saying that, historically, media has focused on individual responsibility for weight, but that in recent years that has been some attention given to the impact of sociocultural factors. However, this comment seems disconnected from the remainder of the Introduction which focuses on how the media promotes weight stigmatizing attitudes.

Throughout the Introduction, please provide a clearer review of previous literature in the area. Have previous studies reviewed the presence of weight bias in British media? Or in the media in other countries? It would be helpful to provide some specific examples that your study builds upon and/or extends.

Relatedly, the summary paragraph on page 3, lines 83-87 should be better developed. In particular, a more detailed summary of related literature, a clear identification of current gaps in knowledge, and how this study will fill those gaps would be helpful.

It would be interesting to incorporate the information on the differences in the news outlets’ characteristics and readerships presented on page 3, lines 103-109 (beginning with “The Mail Online usually supports”) into the Discussion. Could you summarize whether there were any patterns observed across the three news outlets? Did any of the outlets stand out from each other in terms of the content that was published?  

The paragraph starting on page 4, line 134 feels out of place. Can you be clearer with your intention for this paragraph and why you’ve specifically included it here, and also increase clarity/consistency in why you have chosen the language that you have? I agree that this paragraph is important and that we must be intentional with the terminology we choose related to weight and weight stigma – perhaps this information could come earlier in the paper and/or be explicitly linked to a specific portion of the paper (i.e., Results).  

In the Discussion, please further develop the discussion of: 1) this study’s link to previous research, 2) its novelty, and 3) the implication of its novel findings. For instance, the authors note that “media discourses during COVID-19 have taken on particular novel forms” (p. 6, lines 278-279). I do not doubt that this is true – please expand on this idea and discuss specific implications. I am particularly interested in the harm that must stem from the burden of ‘protecting the NHS’, etc. Could you dive into this more?  

Please add a discussion of the limitations of this study.

Minor Comments:

Check grammar throughout the manuscript, for example:

  • Verb tense agreement within sentences (e.g., p. 1, lines 31 & 32, focused v. frames)
  • Provide a specifier with vague pronouns (e.g., “it,” p. 3, line 127)
  • Ensure accuracy within sentences such as, “specific foods are suggested even though they may not be widely available or complicated to make” (p. 4, lines 179-180); for example, this should read “…they may not be widely available or may be complicated to make”

Round 2

Reviewer 1 Report

I appreciate the opportunity to review the revised version of this paper. The authors have thoroughly attended to the feedback provided on the first draft, and the paper is more clear. My only remaining suggestion would be editing for english/academic writing, as currently some of the writing seems more casual in tone. 

Author Response

Response to Reviewer 1 Comments
Point 1: My only remaining suggestion would be editing for english/academic writing, as currently some of the writing seems more casual in tone.
Response 1:
The manuscript has been edited thoroughly edited.
Verbs and sentence structures were edited for academic writing, for example:
ï‚· Page 2, line 44-45 has been edited from ‘The medicalisation of excess weight has led to it being considered a disease...’ to ‘The medicalisation of excess weight has prompted it to be regarded as a disease’.
ï‚· Page 4, line 143-145 has been edited from ‘The media’s role in the production of weight stigma needs to be analysed as a ‘constantly changing social process’, understood in relation to power and social inequality (Parker and Aggleton, 2003, p.14).’ to ‘The production of weight stigma in the media can be understood as a ‘constantly changing social process’, in relation to power and social inequality (Parker and Aggleton, 2003, p.14)’.
Sentences with a casual tone have been identified and rewritten, for example:
ï‚· Page 2, line 62-64 has been edited from ‘Armstrong (1995) argues that public health has increasingly sought to encourage modes of health surveillance that place responsibility on the individual for their health through, for example, emphasis on diet and exercise, as it focuses on risk factors instead of symptoms’ to ‘Public health has increasingly focused on risk factors and encouraged modes of health surveillance that place responsibility on the individual for their health through emphasis on diet and exercise (Armstrong, 1995).’
ï‚· Page 12, line 566-567: has been edited from ‘So, stigma needs to be understood in relation to notions of power and its ability to produce and reproduce social inequalities (Tyler and Slater, 2018)’ to ‘The underlying implication of covering celebrities’ individual stories is that broader societal causes are not discussed (Tyler and Slater, 2018).’

Reviewer 2 Report

While the terminology paragraph (beginning line 70) is now better situated, questions remain regarding the rationale for the use of specific language. The beginning of the paragraph seems to argue against “obesity,” while the middle provides a rationale for not using “fatness.” The authors then state they will use “obesity” in order to avoid perpetuating stigma, though this feels in conflict with the beginning of the paragraph. Can more clear rationale for the authors’ decision be provided?

The sentence on page 3, line 81 (beginning with “The media can play…”) appears to be repeated from the opening line of the Introduction (lines 28-29).

Throughout the Introduction, the additional discussion of previous literature is appreciated. However, the Introduction summary paragraph (p. 3-4, lines 129-150) would benefit from direct identification of gaps that remain in the current knowledge and how this study fills those gaps. Currently, there is a question of how this study builds upon the studies mentioned which investigated weight stigma in the media during COVID-19.

Please continue to revise writing style and grammar throughout manuscript. 

Author Response

Response to Reviewer 2 Comments
Point 1: While the terminology paragraph (beginning line 70) is now better situated, questions remain regarding the rationale for the use of specific language. The beginning of the paragraph seems to argue against “obesity,” while the middle provides a rationale for not using “fatness.” The authors then state they will use “obesity” in order to avoid perpetuating stigma, though this feels in conflict with the beginning of the paragraph. Can more clear rationale for the authors’ decision be provided?
Response 1: The paragraph on the terminology relating to weight (starting on page 3) has been rewritten to provide more clarity. Our aim here is to describe the differing perspectives on what is sensitive language, and how these can be in tension. We suggest first, the reclamation of the word ‘fatness’ as an alternative to ‘person with obesity’, which can be seen to medicalise excess weight and attribute a judgement producing characteristic. In tension with this perspective is how ‘fat’ is regarded as offensive by others. As there is a lack of clear consensus on weight terminology, we set out how we follow the recommendations presented by a UK collaboration. This UK position providers the rationale for the authors decision to use ‘obese’ – despite the risk of medicalising, as above - only when quoting directly from sources of data. We then use person-centred language around obesity and aim to follow an evidence-based approach to avoid assumptions or judgement arising from the text, and so again manage the risk of medicalising experiences.
Point 2: The sentence on page 3, line 81 (beginning with “The media can play…”) appears to be repeated from the opening line of the Introduction (lines 28-29).
Response 2: The repeated sentence has been erased. The paragraph now starts with the following statement from Maiorano et al., (2017) which serves as an introduction to the literature review: ‘The media serves as a platform that can reinforce stigma’ (see page 3, line 114).
Point 3: Throughout the Introduction, the additional discussion of previous literature is appreciated. However, the Introduction summary paragraph (p. 3-4, lines 129-150) would benefit from direct identification of gaps that remain in the current knowledge and how this study fills those gaps. Currently, there is a question of how this study builds upon the studies mentioned which investigated weight stigma in the media during COVID-19.
Response 3: The introduction summary paragraph has been rewritten (page 4-5, line 177-199). Research on obesity discourses during the pandemic is summarised. The following sentences are added to explain how the study fills gaps in the literature: ‘These initial analyses of media coverage during Covid-19 suggest a potential for new meanings and media discourses with impacts on obesity and public health responses. This study aimed to develop closer examination of what these discourses are, as well as identify others.’
Point 4: Please continue to revise writing style and grammar throughout manuscript.
Response 4:
The manuscript has been edited thoroughly edited.
Verbs and sentence structures were edited for academic writing, for example:
ï‚· Page 2, line 44-45 has been edited from ‘The medicalisation of excess weight has led to it being considered a disease...’ to ‘The medicalisation of excess weight has prompted it to be regarded as a disease’.
ï‚· Page 4, line 143-145 has been edited from ‘The media’s role in the production of weight stigma needs to be analysed as a ‘constantly changing social process’, understood in relation to power and social inequality (Parker and Aggleton, 2003, p.14).’ to ‘The production of weight stigma in the media can be understood as a ‘constantly changing social process’, in relation to power and social inequality (Parker and Aggleton, 2003, p.14)’.
Sentences with a casual tone have been identified and rewritten, for example:
ï‚· Page 2, line 62-64 has been edited from ‘Armstrong (1995) argues that public health has increasingly sought to encourage modes of health surveillance that place responsibility on the individual for their health through, for example, emphasis on diet and exercise, as it focuses on risk factors instead of symptoms’ to ‘Public health has increasingly focused on risk factors and encouraged modes of health surveillance that place responsibility on the individual for their health through emphasis on diet and exercise (Armstrong, 1995).’
ï‚· Page 12, line 566-567: has been edited from ‘So, stigma needs to be understood in relation to notions of power and its ability to produce and reproduce social inequalities (Tyler and Slater, 2018)’ to ‘The underlying implication of covering celebrities’ individual stories is that broader societal causes are not discussed (Tyler and Slater, 2018).’

Round 3

Reviewer 2 Report

My only remaining comment is to check for minor errors/typos throughout the document. There are multiple instances of missing punctuation, incorrect punctuation, etc. Please review closely and update.